# Bone Quality in Patients with Parkinson’s Disease Determined by Quantitative Ultrasound (QUS) of the Calcaneus: Influence of Sex Differences

**DOI:** 10.3390/ijerph19052804

**Published:** 2022-02-28

**Authors:** Jordi Caplliure-Llopis, Dolores Escrivá, Esther Navarro-Illana, María Benlloch, Jose Enrique de la Rubia Ortí, Carlos Barrios

**Affiliations:** 1Institute for Research on Musculoskeletal Disorders, Valencia Catholic University San Vincente Martir, Quevedo 2, 46001 Valencia, Spain; jordi.caplliure@ucv.es (J.C.-L.); dolores.escriva@ucv.es (D.E.); carlos.barrios@ucv.es (C.B.); 2Department of Primary Care, Hospital Universitario de La Ribera, 46600 Valencia, Spain; 3Department of Basic Medical Sciences, Catholic University of Valencia San Vicente Martir, 46001 Valencia, Spain; esther.navarro@ucv.es

**Keywords:** Parkinson’s disease, bone quality, quantitative ultrasound, sex

## Abstract

(1) Background: Parkinson’s disease (PD) is a relatively common neurodegenerative disease in elderly individuals, with a high risk of falls. There is abundant literature on the relationship between PD and osteoporosis. The aim of this study is to describe the bone quality of a population with PD by calcaneal ultrasound and to compare it with a healthy control, assessing the influence of possible sex differences. (2) Methods: 21 patients diagnosed with PD were recruited. The control group was composed of 30 healthy individuals with similar sociodemographic characteristics. The bone quality of all participants was assessed using calcaneal quantitative ultrasound (QUS). The parameters recorded were broadband ultrasound attenuation (BUA, in decibels per megahertz), imaging speed of sound (SOS, in meters per second), stiffness index (SI) and T-score of each participant. Bone mineral density (BMD) was estimated using the equation BMD = 0.002592 × (BUA + SOS) − 3.687 (g/cm^2^). (3) Results: significant differences were observed between the healthy control and the PD group: the T-score was lower in the PD group (*p* < 0.05) and SOS was higher in Parkinson’s disease patients (*p* < 0.05), while 28.6% of the PD patients were osteoporotic with T-score values lower than −1.5 compared to 16.7% of osteoporotic individuals in the control group (*p* < 0.01). Regarding the sex, there were significant differences (*p* < 0.05) between the females of the PD group vs. control group, showing a significant difference in the SI (71.4 ± 14.7 vs. 87.8 ± 12), T-score (−2.19 ± 1.1 vs. −0.15 ± 0.8), BUA (104.5 ± 13 vs. 116 ± 10.6) and BMD (0.49 ± 0.09 vs. 0.60 ± 0.08), with no difference in the comparison between the male groups; and the comparison between both sexes in T-score only showed significant differences for the PD group (*p* < 0.05), with worse bone quality in women. (4) Conclusions: this study shows poorer bone quality in female patients with PD, who have a higher percentage of osteoporosis than healthy patients. The QUS technique of the calcaneus seems adequate for these determinations in patients with Parkinson’s disease.

## 1. Introduction

Parkinson’s disease is the most common neurodegenerative pathology among the elderly, with a prevalence in industrialized countries between 0.3% and 1% in the population older than 60 years of age, reaching 3% in people older than 80 [1,2]. In most cases, it is characterized by selective degeneration of dopaminergic neurons in the substantia nigra pars compacta of the midbrain, producing a decreased dopamine transmission throughout the nigrostriatal pathway [3]. In addition, this disease may be familial (associated with an early onset) or sporadic [4,5]. Pathologically, uncontrolled protein aggregation (mainly α-synuclein fibrils), oxidative stress, mitochondrial dysfunction, chronic neuroinflammation (including microglia activation and astrogliosis) and altered autophagy can be observed [6,7,8,9]. It begins with tremor, stiffness and postural instability, progressing towards immobilization, which is why it is considered a highly disabling disease [10,11]. In addition, people who suffer from Parkinson’s disease are at greater risk of falls and fractures due to the symptoms caused by the disease [1,2]. 

Osteoporosis is a skeletal disorder also related to elderly patients, characterized by low bone mass and a deterioration of bone microarchitecture, which results in lower bone quality and increases the risk of fractures [12]. It is an underdiagnosed disease, and many patients with fragility fractures did not have a previous diagnosis of the pathology. In addition to age, other factors, such as sex, steroid use, low body mass index (BMI), sedentary lifestyle, family history, smoking and low levels of vitamin D have been linked to the development of osteoporosis [13]. Among all these, it is interesting to highlight the sex of the individual. In Spain, it is estimated that 2 million women present osteoporosis, compared to 800,000 men. This difference can particularly be seen after the age of 50, with a prevalence of 26.07% in women and 8.1% in men [14].

Some neurological diseases, like epilepsy, multiple sclerosis, dementia and, mainly, Parkinson’s disease, are linked to osteoporosis and a higher risk of fractures [15,16]. In fact, in Parkinson’s disease, osteoporosis is the main cause of fractures [17,18,19]. This could be explained by the high risk of falls compared to the general population, which increases as the disease progresses [13,14], increasing complications due to fractures mainly in the lumbar spine, neck of femur and hip, where there is lower bone mineral density (BMD) [16,18]. 

The sex of the patient with Parkinson’s disease could also be a determining factor for fractures and differences in bone quality, given the differences between both sexes in certain parameters of the pathology. Specifically, the disease is more prevalent in men than in women, especially between 60–69 and 70–79 years of age, respectively [20]. However, there are also differences in the prognosis [21], as it is worse for women, with a higher mortality rate and greater progression of the disease [22], added to the fact that women are at higher risk for falls [23].

Considering the high prevalence of osteoporosis in Parkinson’s disease and the influence of the sex of the individual (dependent variable expressed in years) in the disease, as well as in the osteoporosis (dependent variable determined by QUS), it seems necessary to delve into the analysis of these variables. 

The hypothesis of the study is that the bone status of patients with Parkinson’s disease may be different from that of healthy patients and that the sex of the individual influences this difference. The results of this study may allow us to propose possible therapeutic and preventive interventions linked to bone status. 

BMD is reliably measured using dual-energy X-ray absorptiometry (DXA), a technique used for measuring BMD per area in units of g/cm2. However, it uses ionizing radiation, and the devices used are expensive and impossible to transport, which is a disadvantage in populations with mobility difficulties, such as the elderly or disabled [24]. Moreover, DXA is not available in all hospitals, therefore, limiting its use [25]. The use of QUS of the calcaneus shows a correlation with the DXA method [26,27,28,29,30,31,32]. It has turned out to be a very useful technique to estimate bone quality in a short period of time and is easy to apply and reproduce without any adverse effects [25], which is a good alternative to determine BMD in people with impaired mobility.

Considering the above, the aim of this study was to determine the applicability of the calcaneal QUS to ascertain bone quality and a possible relationship with osteoporosis in patients suffering from Parkinson’s disease, analyzing the influence of sex.

## 2. Materials and Methods

### 2.1. Study Design

A descriptive, quantitative and cross-sectional pilot study was carried out.

### 2.2. Participants

In order to obtain the population sample, Parkinson’s disease associations of Valencia and Castellón (Spain) were contacted, and their members were informed about the nature of the study. The sample was selected from the patients who showed interest in participating and it included patients diagnosed with the disease for more than 6 years and treated with levodopa. Patients with other chronic pathologies which could influence bone status, such as bone cancer, osteomalacia, osteomyelitis, rickets or Paget’s disease, among others, were excluded. A control group of healthy volunteers without any bone pathologies was also recruited. This healthy population was recruited from three neighborhood associations in Valencia. They all voluntarily agreed to be involved in the study after the objectives of the project and procedures were explained to them. 

### 2.3. Procedure

Once the sample was obtained according to the selection criteria, the volunteers and their families received detailed information on the objectives and methodology of the study and signed an informed consent form.

Measurements to determine the bone quality of the study participants were carried out between October and December 2018. Bone mass, weight and height of each individual were measured and recorded using the same equipment, which was regularly calibrated. The body mass index (BMI) was calculated according to the formula BMI = kg/m^2^, where kg is a person’s weight in kilograms and m^2^ is their height in meters squared.

The results were normalized based on age and sex, in the general population by means of the Z-Score, and the comparison was with healthy young people by means of the T-Score, and a range of T-score values was established to define osteoporosis (<−2.5), osteopenia (−1 to −2.5) and normality (>−1).

QUS measurement was used for the evaluation of bone mass parameters using the GE Lunar Achilles Insight (GE Healthcare, Little Chalfont, UK), which is a portable device that allows a quick estimation of 2 basic parameters: the ultrasound broadband attenuation (BUA) and the speed of sound (SOS). BUA refers to the absorption of energy by bone and soft tissue when sound waves travel through them; an increase in BUA correlates with increased bone trabecula content, and the unit is dB/MHz. The SOS parameter refers to the ratio of the length of the body part to the transmission time of the sound waves. Its increase is correlated with reduced bone mineral content. The unit of measurement is meters per second (m/s) [33]. Both ultrasound measurements constitute a clinical variable called the stiffness index (SI) that has been used to determine the risk of osteoporotic fractures and is comparable to BMD measured by the DXA method [34,35]. 

Two ultrasound evaluations of the calcaneus were carried out for each individual, and the mean of both values was later calculated. All ultrasound measurements were performed by the same operator, avoiding bias in the data collection. To obtain bone mass, the manufacturer’s instructions were followed, spraying the calcaneus area with 70° alcohol and placing the foot correctly. Measurements were always made on the nondominant foot.

The intra-operator coefficient of variation for BUA and SOS was 0.26 and 0.18, respectively. According to the criteria for defining bone quality by QUS, T-scores equal to or less than −1.5 were considered an indication of osteoporosis [36]. The estimated heel BMD was calculated using the equation 33 × (BUA + SOS) − 3.6878.

### 2.4. Statistical Analysis

All statistical analyses were performed with the Statistical Package for Social Sciences (SPSS) version 21.0 for Mac (IBM, Chicago, IL, USA). Quantitative data were presented as mean ± standard deviation (SD) and 95% CI. The normality of the distribution of the variables was analyzed with the Kolmogorov–Smirnov test. As some of the parameters did not have a normal distribution, the differences between the subgroups with Parkinson’s disease and the control were analyzed with the non-parametric Mann–Whitney and Wilcoxon tests. Categorical data on the sex of the participants were analyzed with the Chi-square test. Statistical significance was set at *p* < 0.05.

### 2.5. Ethical Considerations

The study was carried out according to the Declaration of Helsinki [37], with prior approval by the Ethics Committee of the Hospital de La Ribera (Valencia, Spain), code 09/072015, acceptance date 15 July 2015.

## 3. Results

After applying the selection criteria described in the previous section, a sample of 21 patients with Parkinson’s disease (Parkinson’s group) and 30 healthy individuals (control group) was obtained. Their sociodemographic characteristics are shown in Table 1. There were no significant differences between both groups in any of the variables. Furthermore, it needs to be highlighted that the males and females of the Parkinson’s disease group had an almost similar mean age (70.7 ± 8.6; versus 71.3 ± 3.7 [z = −0.076; *p* = 0.939]).

Regarding bone quality, the analyzed parameters indicated poorer quality in patients with Parkinson’s disease, with significant differences in the T-score and SOS with lower and higher values, respectively, in this group (Table 2).

Furthermore, 28.6% of the patients with Parkinson’s disease presented T-score values lower than −1.5, considered as indicators of osteoporosis. However, in the control group, only 16.7% of subjects were below −1.5, which represents a significant difference between both groups (*p* < 0.01). When BMD was estimated, 85.7% of patients with Parkinson’s disease had values lower than 0.700 g/cm^2^ that also reflect osteoporosis, compared to 76.7% in the control group, although these differences were not statistically significant (*p* = 0.49).

The sub-analysis by sex revealed that, in the females, patients with Parkinson’s disease showed indicators of poorer bone quality in relation to their peers in the control group, with significant differences in all the parameters analyzed, except for SOS (Table 3). However, in the male participants, no differences were detected between the two groups. Age did not influence these findings because there were no statistically significant differences in the mean age between controls and patients with Parkinson’s disease in both sex groups (males: control group, mean age: 67.9 ± 7.1; Parkinson’s patients: 70.7 ± 8.6 (z = −1.074; *p* = 0.283); females: control group, mean age: 68.5 ± 4.8; patients with Parkinson’s disease: 71.3 ± 3.7 (z = −1.711; *p* = 0.087)).

Finally, when comparing the T-score values between men and women in each of the study groups, significant differences were only observed in the Parkinson group, where women showed poorer bone quality with lower values in the test (Figure 1).

## 4. Discussion

There is a link between the presence of neurological diseases and the development of osteoporosis [15]. Osteoporosis has an impact on osteoporotic fractures, which affect mobility and mortality, and is related to the poor quality of life of the individual [38]. Specifically, in Parkinson’s disease, a high prevalence of osteoporosis has been seen in patients with this pathology [16]. In our study, we can see that the values obtained with the T-Score and SOS tests are in line with osteoporosis prevalence, showing worse bone quality in patients with Parkinson’s disease. From these same indicators, T-Score and BMD, it can also be seen that poor bone quality translates into a higher percentage of osteoporosis in people with Parkinson’s disease compared with healthy people. This may be explained by several factors, highlighting the low level of vitamin D, identified as a risk factor for the disease [39] and associated with a higher risk of falls [40]. 

Immobilization may also be another important factor since, in addition to sunlight deprivation, it increases bone resorption and induces hyperkalemia [41] that inhibits the secretion of parathyroid hormone [42]. Finally, there is the possible influence of Parkinson’s disease medication on bone quality and the appearance of fractures. Antidepressants inhibit serotonin transport systems, damaging the microarchitecture of the bone and reducing BMD [43], and levodopa has side effects, such as hypotension, visual hallucinations and daytime drowsiness, related to an increased risk of falls. Hyperhomocysteinemia also has a direct impact on bone quality [44].

Regarding the variables that may influence the disease, clear epidemiological and clinical differences related to the patients’ sex have been described in Parkinson’s disease. The disease is more common in men, but women have a higher mortality rate accompanied by faster progression and a worse prognosis [22]. In our study, the role of the individual’s sex in the bone quality variable was analyzed, and it can be seen that women with Parkinson’s disease have worse bone quality than healthy women. However, this difference was not observed when comparing the men of both groups (control group and Parkinson’s group), which seems to indicate that the poorer bone quality determined in the Parkinson’s disease group is mainly due to the bone status of the women with the pathology. In addition, these results are in line with those obtained in other studies where the DXA technique was used, and in which worse bone quality was observed in women with Parkinson’s disease compared to healthy women in hips [45], spine [46] and femur [47], which could indicate that the QUS technique used in our study may be useful for patients with Parkinson’s disease. In line with this, when comparing the bone quality of women with men for each of the groups using T-Score, only women in the Parkinson’s disease group showed poorer bone quality, with no differences in the healthy group, despite what is described in the literature for women over 50 years of age [14]. This could be due to homocysteine levels since high amounts of this amino acid in serum have been significantly associated with lower BMD only in women [48]. Moreover, it should also be added that elevated homocysteine levels have been associated with motor alterations, mainly in patients with Parkinson’s disease [49]. The results obtained could also be explained by the influence that body fat has on bone metabolism, having seen that excess weight from an excessive accumulation of fat is related to a decrease in bone mass [50]. In this sense, it has been shown that women with Parkinson’s disease have a greater accumulation of fat (mainly abdominal) than men with the disease [51,52]. In addition, it should be noted that this accumulation is related to the severity of the disease in the early stages [53].

After the analysis and discussion of the study results, a poorer bone quality can be confirmed, as well as a higher percentage of osteoporosis in patients with Parkinson’s disease compared to healthy people. This could be due to the sex of the patients, since women with Parkinson’s disease present worse bone quality than healthy women and these differences cannot be seen in men. To determine these differences, the QUS technique for bone determination of the calcaneus is shown to be effective, which together with the clinical characteristics of patients with Parkinson’s disease and its correlation with DXA densitometry [25,53,54,55], could make it a particularly adequate technique for determination of bone quality in these types of patients.

Regarding the study limitations, it should be noted that bone status prior to Parkinson’s disease diagnosis was unknown. In addition, this group was not stratified by time of incidence of the disease or stages of the disease, as medical records were not accessible. Finally, the small population sample (21 patients with Parkinson’s disease and 30 healthy individuals) was also an important limitation and it would be necessary to replicate the study with a higher number of patients and healthy controls.

On another note, we propose to carry out further studies that allow us to delve into the basic mechanisms underlying our conclusions, where homocysteine levels or the intake of certain drugs could have special relevance.

## 5. Conclusions

The conclusions of the study are poorer bone quality in female patients with Parkinson’s disease, who have a higher percentage of osteoporosis than healthy patients. The QUS technique of the calcaneus seems adequate for these determinations in patients with Parkinson’s disease. At a practical level, the results identify the QUS technique as a suitable method to measure bone density in patients with Parkinson’s disease. Moreover, they seem to indicate that, especially in women, maintaining bone quality could be important in slowing disease progression. Hence, therapeutic measures to promote bone health could be established in this population and implemented in the treatment paradigm. Of note, these measures would need to be applied from the onset of the disease, making it critical to detect signs of osteoporosis as early as possible, for example, using the QUS method. Ultimately, this could help to improve the prognosis, especially for female patients.

## Figures and Tables

**Figure 1 ijerph-19-02804-f001:**
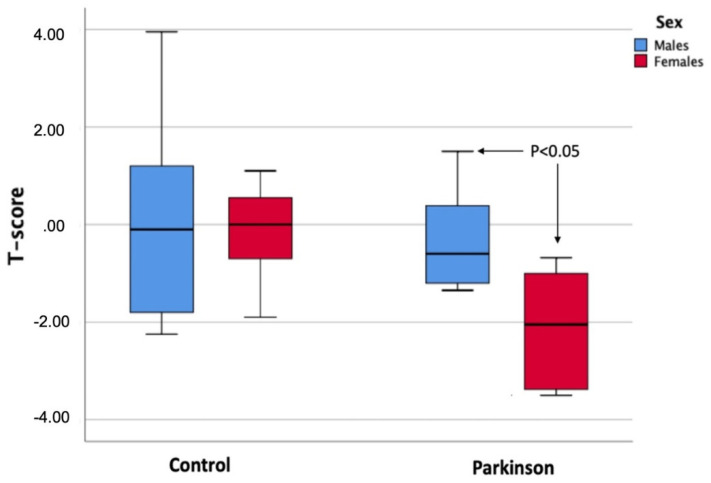
Comparison of T-score values between both sexes in each of the study groups (control group and Parkinson’s group).

**Table 1 ijerph-19-02804-t001:** Sociodemographic characteristics of the study population.

		Parkinson’s Group *N* = 21		Control Group *N* = 30		*p*
Age (Years)		71.75		67.93		0.059
Weight (kg)		73.65		72.20		0.696
Height (cm)		165.05		166.80		0.509
Sex		Frequency	%	Frequency	%	*p*
	Male	12	57.14%	13	43.33%	0.332
	Female	9	42.86%	17	56.67%	

**Table 2 ijerph-19-02804-t002:** Bone parameters related to bone quality between the group of patients with Parkinson’s disease (Parkinson’s group) and healthy people (control group).

	Parkinson’s Group *N* = 21	Control Group *N* = 30	*p*
Stiffness (%)	86.3 ± 20.4	92.1 ± 20.2	0.318
Tscore	−1.05 ± 1.5	−0.12 ± 1.4	0.032 *
BUA (dB/MHz)	115.9 ± 16.5	120.6 ± 15	0.304
SOS (m/s)	1530.3 ± 42	1164 ± 646.3	0.013 *
BMD (gr/cm^2^)	0.58 ± 0.13	0.62 ± 0.14	0.283

BMD: Bone mineral density. BUA: broadband ultrasound attenuation. SOS: speed of sound. *: Statistically significant differences *p* < 0.05.

**Table 3 ijerph-19-02804-t003:** Sex differences in bone quality parameters for each of the study groups (Parkinson’s and healthy control).

	Male			Female		
	Parkinson’s Group *N* = 12	Control Group *N* = 13	*p*	Parkinson’s Group *N* = 9	Control Group *N* = 17	*p*
Stiffness (%)	97.4 ± 4	97.3 ± 27.2	0.977	71.4 ± 14.7	87.8 ± 12	0.005 *
Tscore	−0.20 ± 1.3	−0.08 ± 1.9	0.863	−2.19 ± 1.1	−0.15 ± 0.8	0.001 *
BUA (dB/MHz)	124.5 ± 13.8	126.5 ± 18	0.760	104.5 ± 13	116 ± 10.6	0.023 *
SOS (m/s)	1546.9 ± 42.4	1333.9 ± 549.1	0.194	1508.2 ± 31.4	1034.2 ± 699.8	0.056
BMD (mg/cm^2^)	0.64 ± 0.1	0.65 ± 0.1	0.895	0.49 ± 0.09	0.60 ± 0.08	0.007 *

BMD: Bone mineral density. BUA: broadband ultrasound attenuation. SOS: speed of sound. *: Statistically significant differences *p* < 0.05.

## Data Availability

Not applicable.

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
