# Peer review of "Bone Quality in Patients with Parkinson’s Disease Determined by Quantitative Ultrasound (QUS) of the Calcaneus: Influence of Sex Differences"

_ijerph, 2022, doi:10.3390/ijerph19052804_

Round 1
Reviewer 1 Report
Parkinson’s disease (PD) is among the most common neurodegenerative diseases. Affected patients typically develop motor symptoms such as postural instability and tremor resulting in increased risk for falls and bone fractures. Of note, osteoporosis, which also increases the risk of fractures is another disorder frequently observed in the elderly, particularly in women. Previous studies revealed that PD patients have a higher risk of osteoporosis and low bone mineral density compared to healthy controls. In the manuscript entitled “Bone quality in patients with Parkinson’s determined by quantitative ultrasound (QUS) of the calcaneus. Influence of sex differences.” Caplliure-Llopis et al., report their efforts to determine bone quality of PD patients by calcaneus ultrasound and evaluate differences between male and female patients. In accordance with previous studies, they observed significant differences in bone health between PD patients and healthy controls, which were particularly present in the female cohort. Considering the major disadvantages of bone density measurements with conventional methods, it could be beneficial to develop alternative strategies to assess bone health. Thus, this pilot study provides evidence for QUS being an adequate system to screen PD patients for osteoporosis.
Broad comments
Overall, the study results are clearly presented, and the manuscript is well organized. Worth mentioning is also the detailed explanation of used parameters in the Materials and Methods section. However, there are several points of concern listed in the following:
1) The manuscript was submitted to be published in the special issue “Promotion of Health Habits to Prevent and Treat Mental and Neurological Disorders”. This issue specifically focuses on “the importance of the promotion of health habits, for example, through a nutrition and physical activity intervention, in the prevention and treatment of different neurological disorders”. However, in the present state, reported findings are not obviously related to this topic. Therefore, it would be beneficial, if the authors could expand more on how their study fits into the subject matter of this special issue.
2) Previous studies have shown that lower bone mass is directly associated with increased disease severity. It would be a major benefit if the present study could comment on this association as well. However, as indicated in the discussion, relevant medical records were not accessible.
3) The tested cohort (21 PD patients and 30 controls) is quite small. Given that the study reproduced previous findings, it seems that this is not a major concern. However, it would be good to list this as a limitation of the study.
4) Due to grammatical issues or incorrect word choice, some sentences are hard to understand. Thus, to make it easier for the reader to follow the explanations, minor language editing would be beneficial. Please find some examples and suggestions in the specific comments.
Specific comments
1) To be scientifically correct, it is important to use complete medical terms such as “Parkinson’s disease” instead of “Parkinson’s”. This especially applies to the title. Thus, it would be beneficial to replace “Parkinson’s” by “Parkinson’s disease” when applicable, for example: title, line 37, 61, 83, 86, 91…
2) The introduction provides only a very short description on the background of Parkinson’s disease. To make it easier for readers to follow, I would recommend adding one or two sentences, for example explaining pathomechanisms of PD (loss of dopaminergic neurons in the substantia nigra).
3) Materials and Methods:
3.1 In section 2.2, the recruitment procedures for patients and inclusion criteria are listed. Similarly, it would be advantageous to mention the specificities of the control recruitment process and exclusion criteria.
3.2 line 105-108: T-score values are supposed to be negative values, please correct
4) Word choice:
4.1 line 50: the term“On the other hand” does not fit in this context.
4.2 line 72: replace ”woman are more at risk” by “woman are at higher risk”
4.3 line 78: replace “…either, making its use limited” by “therefore limiting its use”
4.4 line 97: replace “after” by “according to”
4.5 line 219: choose other word for “unwell”
4.6 line 221: choose other word for “sick”
5) Some sentences would benefit from revision to make it easier for the reader to follow
5.1 line 57-58: …”it is interesting to highlight the sex of the individual, since in Spain”…
5.2 line 123-124: replace …“and correct food placement” with “placing the foot correctly”
6) Grammar/spelling errors:
6.1 line 145: replace “there were not” with “there were no”
6.2 line 154: add commas before and after “respectively”
6.3 Table 3: replace “stifness” by “stiffness”
6.4 line 248: remove “of” after “make”
Reviewer 2 Report
Introduction: Please provide the research hypothesis. Please specify the dependent and independent variables and the indicators of these variables.
Materials and Methods: Please describe in detail the method of selection for testing. Please provide the number and date of the ethics committee.
Discussion:
The discussion needs to be expanded. Please see other studies, for example: Cereda E, Cassani E, Barichella M, Caccialanza R, Pezzoli G. Anthropometric indices of fat distribution and cardiometabolic risk in Parkinson’s disease. Nutr Metab Cardiovasc Dis 2013; 23: 264-271.
WilczyĹ„ski J, PóĹ‚rola P. Body composition assessment by bioelectrical impedance analysis among patients treated with levodopa for Parkinson’s disease e. Medical Studies 2018; 34 (2): 1–7) https://doi.org/10.5114/ms.2018.76872.
Vikdahl M, Carlsson M, Linder J, Forsgren L, Håglin L. Weight gain and increased central obesity in the early phase of Parkinson’s disease. Clin Nutr 2014; 33: 1132-1139.
Conclusions: Please indicate the practical importance of these studies.
Reviewer 3 Report
Please see comments in the attached pdf file.

Author Response
Thank you very much for your contributions. We believe we have changed and completed what has been indicated by the reviewer.

Reviewer 4 Report
Osteoporosis is frequently seen in Parkinson’s disease (PD). This study reconfirms the association between PD and osteoporosis. This manuscript also finds that female PD patients have higher bone abnormality than male PD patients.
Here are my suggestions and comments to further improve the manuscript, which authors may wish to consider.
- There is a significant difference between the age of PD patients and controls. Have the authors confirmed that the age difference does not affect their finding and analysis.
- Also, because authors compared separately female PD patients and female controls and found a significant difference in bone health parameters. A statistical analysis of age of this subgroup should be performed to rule out the influence of aging on their findings.
- As also noted by authors in the discussion that duration of disease could affect the bone density differentially. Compared to control the bone parameters were almost similar in male PD patients. Only female patients showed significant changes. Though it is difficult to find out the disease duration, how about comparing the age differences between male PD and female PD patients ? This should be done.
- Table 3, typo in P value column should be corrected (0.000*).
Round 2
Reviewer 1 Report
Thank you for carefully revising your manuscript. Please find my comments in the attached document.
